# Endothelial Lipase Modulates Paraoxonase 1 Content and Arylesterase Activity of HDL

**DOI:** 10.3390/ijms22020719

**Published:** 2021-01-13

**Authors:** Irene Schilcher, Julia T. Stadler, Margarete Lechleitner, Andelko Hrzenjak, Andrea Berghold, Gudrun Pregartner, Marie Lhomme, Michael Holzer, Melanie Korbelius, Florian Reichmann, Anna Springer, Christian Wadsack, Tobias Madl, Dagmar Kratky, Anatol Kontush, Gunther Marsche, Saša Frank

**Affiliations:** 1Gottfried Schatz Research Center for Cell Signaling, Metabolism and Aging, Molecular Biology and Biochemistry, Medical University of Graz, Neue Stiftingtalstraße 6/6, 8010 Graz, Austria; irene.schilcher@medunigraz.at (I.S.); margarete.lechleitner@medunigraz.at (M.L.); m.korbelius@medunigraz.at (M.K.); anna.springer@medunigraz.at (A.S.); tobias.madl@medunigraz.at (T.M.); dagmar.kratky@medunigraz.at (D.K.); 2Otto Loewi Research Center, Division of Pharmacology, Medical University of Graz, Universitätsplatz 4, 8010 Graz, Austria; julia.stadler@medunigraz.at (J.T.S.); michael.holzer@medunigraz.at (M.H.); florian.reichmann@medunigraz.at (F.R.); gunther.marsche@medunigraz.at (G.M.); 3Division of Pulmonology, Department of Internal Medicine, Medical University of Graz, Auenbruggerplatz 16, 8036 Graz, Austria; andelko.hrzenjak@medunugraz.at; 4Ludwig Boltzmann Institute for Lung Vascular Research, Stiftingtalstrasse 24, 8010 Graz, Austria; 5Institute for Medical Informatics, Statistics and Documentation, Medical University of Graz, Auenbruggerplatz 2, 8036 Graz, Austria; andrea.berghold@medunigraz.at (A.B.); gudrun.pregartner@medunigraz.at (G.P.); 6ICANalytics Lipidomics, Institute of Cardiometabolism and Nutrition, 75013 Paris, France; m.lhomme@ican-institute.org; 7Department of Obstetrics and Gynecology, Medical University of Graz, Auenbruggerplatz 14, 8036 Graz, Austria; christian.wadsack@medunigraz.at; 8BioTechMed-Graz, Mozartgasse 12/II, 8010 Graz, Austria; 9INSERM Research Unit 1166—ICAN, Sorbonne University, 75013 Paris, France; anatol.kontush@upmc.fr

**Keywords:** endothelial lipase, high-density lipoprotein, paraoxonase 1, arylesterase activity, mass spectrometry, NMR spectroscopy

## Abstract

Endothelial lipase (EL) is a strong modulator of the high-density lipoprotein (HDL) structure, composition, and function. Here, we examined the impact of EL on HDL paraoxonase 1 (PON1) content and arylesterase (AE) activity in vitro and in vivo. The incubation of HDL with EL-overexpressing HepG2 cells decreased HDL size, PON1 content, and AE activity. The EL modification of HDL did not diminish the capacity of HDL to associate with PON1 when EL-modified HDL was incubated with PON1-overexpressing cells. The overexpression of EL in mice significantly decreased HDL serum levels but unexpectedly increased HDL PON1 content and HDL AE activity. Enzymatically inactive EL had no effect on the PON1 content of HDL in mice. In healthy subjects, EL serum levels were not significantly correlated with HDL levels. However, HDL PON1 content was positively associated with EL serum levels. The EL-induced changes in the HDL-lipid composition were not linked to the HDL PON1 content. We conclude that primarily, the interaction of enzymatically active EL with HDL, rather than EL-induced alterations in HDL size and composition, causes PON1 displacement from HDL in vitro. In vivo, the EL-mediated reduction of HDL serum levels and the consequently increased PON1-to-HDL ratio in serum increase HDL PON1 content and AE activity in mice. In humans, additional mechanisms appear to underlie the association of EL serum levels and HDL PON1 content.

## 1. Introduction

The atheroprotective effects of serum high-density lipoprotein (HDL) are largely ascribed to its anti-oxidative, anti-inflammatory, cholesterol efflux, and endothelial function, which partially rely on HDL-associated paraoxonase 1 (PON1) [1,2,3,4,5]. PON1 is a serum enzyme synthetized by the liver. Binding of the secreted PON1 to HDL involves an interaction of HDL with the hepatocyte surface, which is a process facilitated by scavenger receptor class B type I (SR-BI) [6,7]. The protein–protein and protein–lipid interactions between PON1 and HDL have been found to be crucial for the stability and enzymatic activity of HDL-associated PON1 [8,9,10]. In addition to its HDL-associated form, PON1 exists also as a free enzyme, which can redistribute to cell membranes or can be taken up by cells [11,12]. 

Although the majority (95%) of secreted PON1 is associated with HDL, only a subset of circulating HDL particles (5–10%) contain PON1 [13,14,15,16]. It has been shown that the lipid and protein composition of PON1-containing HDL particles differ from HDL particles lacking PON1 [17]. 

In humans, a strong negative correlation between PON1 activity and serum concentrations of lipid oxidation markers was observed [18,19]. Similarly, PON1 deficiency in mice is associated with increased oxidative stress [20,21], and decreased oxidative stress is observed upon PON1 overexpression [22].

Endothelial lipase (EL) is a phospholipase expressed primarily by vascular endothelial cells, vascular smooth muscle cells, and macrophages [23,24]. Based on increased HDL serum levels in EL-deficient mice as well as in humans expressing EL variants with decreased enzymatic activity, EL emerged as a strong negative regulator of HDL plasma levels [25,26]. By hydrolyzing HDL, EL depletes primarily phospholipids generating lysophospholipids and free fatty acids, which partially remain associated with EL-modified HDL and partially distribute to albumin as well as underlying EL-expressing cells [27,28]. EL modification of HDL markedly alters the structural and functional properties of HDL [27,29,30,31]. EL modulates also plasma levels of apolipoprotein B-containing lipoproteins in mice and is associated with a proatherogenic lipid profile and subclinical atherosclerosis in humans [32,33,34,35]. While studies in mice generated inconclusive data regarding the role of EL in atherosclerosis [36,37], hepatic EL overexpression decreased cholesterol diet-induced hypercholesterolemia and atherosclerosis in transgenic rabbits [38]. 

HDL-associated PON1 content and activity rely on the interaction of PON1 with HDL phospholipids, cholesterol, and apolipoprotein A-I (apoA-I) [8,9]. Considering the pronounced impact of EL on structural and compositional features of HDL [27,30], it is conceivable that the interaction and association of PON1 with HDL and in turn the PON1 content and arylesterase (AE) activity of HDL are altered by EL. Indeed, in our recent study, in vitro EL modification of HDL decreased HDL PON1 content [30]. To study this phenomenon in more detail, in the present study, we conducted a more in-depth study in vitro and additionally investigated the relationship between EL and HDL PON1 content as well as AE activity in mice and humans in vivo. 

## 2. Results

### 2.1. EL Decreases HDL-Associated PON1 Content and AE Activity In Vitro 

To examine the impact of EL on HDL-associated PON1, purified human HDL was incubated with EL-overexpressing or empty virus (EV)-infected control HepG2 cells in the absence or presence of PON1 co-overexpression. Following the isolation of modified HDL by ultracentrifugation, both the PON1 content (Figure 1A) and AE activity (Figure 1B) were significantly lower in EL-HDL compared to EV-HDL. This was irrespective of whether HDL modification was performed in the absence or presence of PON1 co-overexpression. Similar results were obtained with EL-HDL and EV-HDL generated in the presence of 4% bovine serum albumin (BSA) to prevent the accumulation of lipolytic products in HDL [30] (Appendix A). The decreased PON1 content of EL-HDL compared to EV-HDL was also observed when aliquots of the incubation media were analyzed by native gel electrophoresis followed by PON1 Western blotting (Figure 1C). Therefore, ultracentrifugation as a possible cause for the decreased PON1 content of EL-HDL can be excluded. In sharp contrast to the effects of enzymatically active EL, the incubation of HDL with HepG2 cells overexpressing enzymatically inactive EL (M-EL) resulted in significantly increased PON1 content (Figure 1D) but did not alter significantly the AE activity of M-EL-HDL (Figure 1E). In the presence of PON1 co-overexpression, EV-HDL and M-EL-HDL had comparable PON1 content (Figure 1D) and AE activity (Figure 1E). These results clearly suggest that the lipolytic activity of EL is crucial for decreasing HDL PON1 content and AE activity.

### 2.2. EL Decreases HDL Size and PON1 Content without Diminishing the Capacity of HDL to Associate with PON1 

Since EL-induced HDL size reduction may explain PON1 displacement from HDL, we studied the relationship between HDL size and PON1 content. We analyzed the PON1 content and AE activity of EV-HDL and EL-HDL subpopulations with a defined size prepared by fast protein liquid chromatography (FPLC) (Appendix A). As shown in Figure 2A,B, the decrease in particle size of both types of modified HDL was associated with a decrease in PON1 content and AE activity. These results strongly suggest the involvement of an EL-induced reduction of HDL size in diminishing HDL PON1 content and AE activity. However, the EL-induced HDL size and PON1 reduction did not diminish the capacity of EL-modified HDL to associate with PON1 when incubated with PON1-overexpressing cells (Figure 2C,D). Therefore, we concluded that primarily the interaction of enzymatically active EL with HDL, rather than EL-induced alterations in HDL size and composition, causes PON1 displacement from HDL. 

### 2.3. EL Overexpression in Mice Increases HDL PON1 Content and AE Activity 

To examine whether EL overexpression in vivo affects HDL PON1 content and AE activity, we analyzed PON1 and apoA-I levels as well as AE activity of serum of mice injected with EL-Ad or control EV-Ad. While apoA-I levels were below the detection limit, the PON1 levels were similar in EL serum and apoB-DS compared to the respective EV controls (Figure 3A). Interestingly, despite comparable PON1 levels, the AE activity of EL serum and apolipoprotein B-depleted serum (apoB-DS) was significantly lower compared to the respective EV controls (Figure 3B).

Unaltered PON1 serum levels despite the markedly decreased HDL serum levels in EL-overexpressing mice may suggest either an HDL overload with PON1 or an accumulation of PON1 that is not associated with HDL. To clarify this, we analyzed the PON1 content of HDL isolated from EV and EL mouse serum by FPLC. In line with markedly decreased HDL levels in EL compared to EV serum (Figure 3C), apoA-I levels in the EL-FPLC fractions 30–36 (corresponding to HDL) were drastically reduced, but PON1 levels were comparable when compared to the corresponding EV-FPLC fractions (Figure 3D). These findings suggested an EL-overexpression-driven overload of HDL with PON1. Indeed, mouse EL-HDL isolated by ultracentrifugation exhibited a 4.3-fold higher PON1 content and 1.6-fold higher AE activity when compared to control EV-HDL (Figure 3E,F). These in vivo findings led us to the conclusion that a marked EL-induced decrease in HDL serum levels increases the PON1-to-HDL ratio, leading in turn to a marked PON1 overload of HDL, which is accompanied by less profoundly increased PON1 AE activity.

### 2.4. HDL PON1 Content Is Positively Associated with EL Serum Levels in Humans

Next, we examined the relationship between EL serum levels, HDL PON1 content, and AE activity in humans. For this purpose, we recruited healthy volunteers (baseline characteristics are shown in Appendix A)OK and measured EL serum levels as well as PON1 content and AE activity of isolated HDL. We found significantly higher PON1 content in HDL isolated from subjects with high (above median) compared to those with low (below median) EL serum levels (Figure 4A) and EL serum levels were significantly positively correlated with HDL PON1 content (Figure 4B). The HDL PON1 content was strongly correlated with HDL AE activity (r = 0.52, *p* < 0.001). However, HDL AE activity showed only a trend to be increased in subjects with above median EL expression (Figure 4C) and was not significantly correlated with EL serum levels (Figure 4D).

### 2.5. EL-Induced Alterations of HDL Lipidome and HDL Serum Levels Are Not Linked to HDL PON1 Content 

We hypothesized that the positive association between EL serum levels and HDL PON1 content in humans reflects EL-induced alterations in the HDL lipid composition, which facilitates the loading of HDL with PON1. To test this hypothesis, we performed logistic regression analysis to identify HDL lipid species significantly associated with high PON1 content and then additionally adjusted the model for EL (low vs. high). We identified 16 lipid species that were significantly positively or negatively associated with high HDL PON1 content (Appendix A). The levels of these species in samples containing high or low PON1 concentrations are shown in Appendix A. However, after adjusting for EL (high vs. low), the majority of the associations (with the exception of lysophosphatidylcholine (LPC) 22:4, phosphatidic acid (PA) 32:1, phosphatidylinositol (PI) 38:4, ceramide (Cer) d18:2/22:0, Cer d18:2/24:0, and Cer d18:2/26:0) remained significant (Appendix A). In line with this, HDL particles that differ in lipid composition dependent on EL serum levels (low or high) had comparable content of PON1 (Appendix A). These data clearly suggested that the association between the PON1 content and lipid composition of HDL is not significantly affected by EL. According to this, the observed positive relationship between EL serum levels and HDL PON1 content (Figure 4) cannot be explained by EL-induced alterations in the HDL lipid composition. 

We observed that the EL-induced reduction of HDL levels was accompanied by increased HDL PON1 content in EL-overexpressing mice. Next, we assessed whether EL serum levels in humans are linked to serum concentrations of HDL parameters quantified by nuclear magnetic resonance (NMR) spectroscopy. Serum levels of the HDL parameters including total HDL as well as HDL subclasses quantified on the basis of their cholesterol, triacylglycerol, phospholipid, apoA-I or apoA-II content were comparable in the low and high EL sera (Appendix A) and were not significantly correlated with EL (Appendix A). Accordingly, the positive association between EL serum levels and HDL PON1 content cannot be directly attributed to the EL-induced HDL reduction and the concomitantly increased PON1-to-HDL ratio, as found in EL-overexpressing mice.

## 3. Discussion

In the present study, we provide evidence that EL modulates HDL PON1 content and AE activity and that marked differences are observed in vitro and in vivo. 

In vivo, the concerted action of different enzymes and receptors as well as interactions of HDL with other lipoproteins modulate the HDL size and composition [1]. In our in vitro model, HDL remodeling is primarily due to EL and, to a lesser extent, the interaction of HDL with HepG2 cells as well as enzymes and lipids secreted by HepG2 cells [14]. However, EL-modified HDL generated by the incubation of human plasma with EL-overexpressing cells also exhibited decreased PON1 content and AE activity (Appendix A). This finding indicates that the diminishing effect of EL on HDL PON1 content and AE activity in vitro was also operative under more physiological conditions and was not counteracted by the HDL remodeling machinery of human plasma. 

It is well established that HDL phospholipids play an important role in the association of PON1 with HDL [9,10] and that EL by its profound phospholipase activity efficiently degrades HDL phospholipids [28]. Accordingly, the EL-mediated alterations in the HDL phospholipid abundance and composition [27,30] are likely the cause for the decreased PON1 content and AE activity in the in vitro EL-modified HDL. 

Previous studies have shown that PON1-containing HDL particles contain decreased levels of unsaturated LPC species [17]. The fact that EL generates primarily unsaturated LPC species by cleaving HDL phospholipids [28,39] suggest that the accumulation of these EL-generated LPCs contributes to the reduction of PON1 content in EL-HDL. However, EL efficiently decreased HDL PON1 also in the presence of BSA (Appendix A), which binds LPC and prevents their accumulation in EL-modified HDL [30]. Additionally, the observation that human HDL particles with high PON1 content contain higher levels of LPC 22:4 compared to those with low PON1 (Appendix A) supports the notion that EL-generated LPCs are not a likely cause for PON1 reduction in EL-modified HDL. 

A positive relationship between HDL size and PON1 content, revealed by PON1 quantification in FPLC-generated HDL fractions of various size, implicated EL-mediated HDL size reduction in diminishing HDL PON1 content. The importance of the enzymatic activity of EL in displacing PON1 from HDL in vitro was also demonstrated by the increased, rather than decreased, PON1 content of HDL following the incubation of HDL with cells expressing enzymatically inactive EL. This most probably reflects EL-mediated bridging of HDL to the cells, which in turn facilitates HDL-mediated desorption of the secreted PON1 [6], as described for SR-BI [7]. 

Importantly, the EL-induced alterations in HDL size and composition, accompanied by a displacement of PON1, did not diminish the capacity of the EL-modified HDL to associate with PON1. Accordingly, it seems that the interaction of enzymatically active EL with HDL and/or an active EL-mediated lipolysis of HDL, rather than altered HDL size and composition, might be crucial for PON1 displacement from HDL. 

In sharp contrast to the EL-mediated HDL PON1 reduction observed in vitro, EL overexpression in mice in vivo increased the PON1 content of HDL. It is conceivable that a massive EL-induced HDL reduction and a concomitantly increased serum PON1-to-HDL ratio drive the overload of HDL with PON1. The findings that endogenous PON1 expression was not increased by EL overexpression (Appendix A), and that the overexpression of enzymatically inactive EL did not increase HDL PON1 content (Appendix A), exclude the possibility that an increased endogenous PON1 synthesis or EL bridging function contributed to the HDL PON1 overload in EL-overexpressing mice. Moreover, comparable hepatic apoA-I mRNA levels in EL-overexpressing and control mice (Appendix A) strongly argue against attenuated apoA-I synthesis as the cause of EL-induced HDL reduction. Previous studies have also shown that apoA-I synthesis is not affected by adenovirus-induced EL overexpression in mice and, importantly, that the reduced HDL serum concentrations are exclusively due to the EL-mediated acceleration of HDL catabolism [40,41]. Additionally, structural and compositional features of the in vivo EL-modified mouse HDL, including unaltered particle size, increased phospholipid and triacylglycerol as well as decreased lysophospholipid content (in contrast to reduced particle size, decreased phospholipid and triacylglycerol as well as increased lysophospholipid content observed in the in vitro EL-modified human EL-HDL) [27], may have facilitated the PON1 overload of the in vivo EL-modified mouse HDL. 

Interestingly, however, the increased PON1 content of the EL-modified mouse HDL (4.3-fold higher compared to control mouse EV-HDL) was accompanied by only 1.6-fold increased AE activity. Considering that approximately only 5–10% of HDL particles in normal serum contain PON1 [13,14], the increased PON1 content of the in vivo EL-modified mouse HDL reflects either an increased number of PON1-containing HDL particles or an increased number of PON1 molecules per HDL particle. Accordingly, the observed discrepancy between the PON1 content and AE activity might be due to the EL-overexpression-driven association of PON1 with a subset of HDL particles, which under normal conditions would not contain PON1 and whose structure and composition do not provide a suitable micro-environment to support PON1 AE activity. Alternatively, the EL-overexpression-driven increase in PON1 load per HDL particle could preclude an appropriate interaction of PON1 with HDL lipids and proteins [8,9], resulting in an altered PON1 conformation and in turn decreased AE activity. Moreover, the decreased LPC levels in the in vivo EL-modified mouse HDL [27] may, considering the stimulating effect of LPC on PON1 AE activity [42], also, at least in part, be responsible for the decreased AE activity. However, in human HDL samples, AE activity was associated with several HDL lipid species (Appendix A) but not LPC (not shown). 

Previous studies reported a negative correlation between EL and HDL serum levels in patients with cardiovascular disease but not in those without [35,43]. In line, the EL serum levels in the present study were not significantly correlated with either parameter indicative of HDL serum bioavailability measured by NMR spectroscopy (Appendix A). Nevertheless, significantly higher levels of LPC and LPE species (lipolytic products known to be generated by EL-mediated phospholipolysis [27,28,30]) in HDL isolated from subjects with high compared to low EL serum (Appendix A) strongly argue for the modulation of the HDL composition by EL in vivo. This is further supported by a significant positive association of the LPC and LPE species with high EL serum levels (Appendix A) However, results of the logistic regression analysis (Appendix A) and the finding that HDL particles which differ in lipid composition, dependent on whether they originate from low or high EL serum, have comparable (high or low) content of PON1 (Appendix A) suggest that in humans, the EL-related lipidomic signature of HDL is not a determinant of the HDL PON1 content. 

What are the functional implications of EL-induced changes in HDL PON1 content? In line with a role of PON1 in the cholesterol efflux capacity of HDL [5,44], in the present study, we found a robust positive correlation between the cholesterol efflux capacity of HDL isolated from healthy subjects and the HDL PON1 content and AE activity, respectively (r = 0.4, *p* = 0.003 and r = 0.53, *p* < 0.001, respectively). Accordingly, the decreased cholesterol efflux capacity of the in vitro EL-modified HDL described in our previous report [27] was probably (at least in part) due to the decreased HDL PON1 content. In the same study [27], the cholesterol efflux capacity of the in vivo EL-modified mouse HDL was not altered, despite a PON1 overload. This indicates that physiological levels and/or the distribution of PON1 on the subset of HDL particles that provide an optimal micro-environment are crucial for the exertion of maximal biological PON1 activity. 

In a previous study, we found an increased antioxidative capacity of in vitro EL-modified human HDL despite decreased PON1 content [30], disputing the antioxidative role of PON1. However, it is possible that the decreased HDL PON1 was compensated by the profound antioxidative activity of LPCs enriched in HDL [30]. In contrast to the in vitro EL-induced increase in the antioxidative capacity of HDL, the in vivo EL-modified mouse HDL overloaded with PON1 exhibited a strong prooxidative activity, which was comparable to control HDL (unpublished results). It is likely that the systemic inflammatory response induced by adenoviral transduction produced dysfunctional HDL, highlighting again the failure of PON1 overload to entail functional benefit. Further studies are warranted to understand the complex relationship between the antioxidative capacity of HDL and the HDL PON1 content, AE activity, and lipid composition in humans.

We demonstrated that EL profoundly modulates HDL PON1 content and AE activity in vitro and in vivo. We conclude that primarily the interaction of enzymatically active EL with HDL, rather than EL-induced alterations in HDL size and composition, causes PON1 displacement from HDL in vitro. In vivo, the EL-mediated reduction of HDL serum levels and consequently increased serum PON1-to-HDL ratio increase HDL PON1 content and AE activity in mice. In humans, additional mechanisms appear to underlie the association of EL serum levels and HDL PON1 content. Therefore, further larger studies are needed to understand the role of EL in modulation of HDL PON1 content in humans. 

## 4. Materials and Methods

### 4.1. Cell Culture

HepG2 cells (ATCC^®^ HB-8065^TM^, Manassas, VA, USA) were cultured as described [27].

### 4.2. Human Plasma

Normolipidemic plasma of 18 healthy donors was obtained as described [30].

### 4.3. Blood Collection from Healthy Volunteers 

Fasting blood samples were collected by antecubital venipuncture into two 9 mL serum tubes (VACUETTE TUBE 9 mL Z Serum Clot Activator). Blood was incubated at 20 °C for 30 min followed by centrifugation at 1800 should be like this, please do not change× *g* at 20 °C for 10 min. Serum aliquots were frozen at −80 °C. The study was approved by the Ethics Committee of the Medical University of Graz (32-096 ex 19/20), and informed consent of all probands was obtained in accordance with the Declaration of Helsinki.

### 4.4. Isolation of HDL 

HDL from human plasma or serum as well as mouse serum or cell culture media was prepared by a one-step density gradient ultracentrifugation as described [27]. 

### 4.5. Modification of HDL and Plasma

HDL and plasma were incubated with HepG2 cells overexpressing human EL or enzymatically inactive human EL or with control cells infected with empty adenovirus containing no recombinant cDNA as described [27,30]. Some modifications were performed in the presence of 4% (final concentration) of non-esterified fatty acid (NEFA)-free bovine serum albumin (BSA) (Sigma-Aldrich, Vienna, Austria) or PON1 co-overexpression achieved by co-infection of the cells with multiplicity of infection (MOI) 20 of PON1-adenovirus (Vector Biosystems Inc., Malvern, PA, USA). Modified HDLs were re-isolated and stored as described [27,30].

### 4.6. Preparation of In Vivo EL-Modified Serum and Control EV Serum

Modified serum was isolated from 12-week-old male C57BL/6J mice injected with EL-Ad or EV-Ad as described in our previous report [27]. The serum was used for the preparation of apoB-depleted serum (apoB-DS) [27] as well as for HDL isolation by ultracentrifugation or by fast protein liquid chromatography (FPLC). All experimental protocols related to animal experiments were approved by the Austrian Federal Ministry of Education, Science and Research (BMWF-66.010/0020-WF/V/3b/2016). All experiments were performed in accordance with relevant guidelines and regulations.

### 4.7. Fractionation of HDL by FPLC 

HDL samples were loaded on a NGC QUEST FPLC System (Bio-Rad, Vienna, Austria) equipped with a Superdex 200 Increase 10/300 column (GE Healthcare Europe GmbH, Munich, Germany) using phosphate buffered saline (PBS) as running buffer. HDL fractionation was started after 18 min with a constant flow of 0.5 mL per fraction.

### 4.8. FPLC of Mouse Serum 

A pool of 200 µL of mouse serum of EV- or EL-overexpressing mice was subjected to FPLC on a Pharmacia FPLC system (Pfizer Pharma, Karlsruhe, Germany) equipped with a Superose 6 column (Amersham Biosciences, Piscataway, NJ, USA). OK Lipoproteins were eluted with 10 mmol/L Tris-HCl, 1 mmol/L ethylenediaminetetraacetic acid (EDTA), 0.9% NaCl, and 0.02% NaN_3_ (pH 7.4). The total cholesterol (TC) (Greiner Diagnostics AG, Bahlingen, Germany) concentrations in 0.5 mL fractions were determined spectrophotometrically.

### 4.9. SDS-PAGE and Western Blotting

HDL-associated proteins in isolated HDL (10 µg), FPLC fractions (10 µL), serum (1.5 µL), or apoB-DS (2 µL) were quantified using antibodies specific for human/mouse PON1 (abcam-ab24261, dilution 1:1000, Cambridge, UK), human apoA-I (Novus biological, NB100-65491, dilution 1:3000, Littleton, CO, USA), or mouse apoA-I (Santa Cruz Biotechnology, sc-30089, LOT D2913, dilution 1:500, Heidelberg, Germany) as described [27,30]. 

### 4.10. Non-Denaturing Gradient-Gel Electrophoresis and Western Blotting

Modified HDL (10 µg), FPLC fractions (10 µL), or incubation media (containing 15 µg protein) were electrophoresed on 4–20% non-denaturing polyacrylamide gels as described [27]. For the analyses of PON1 content, separated proteins were transferred to polyvinylidene difluoride (PVDF) membranes (Carl Roth, Karlsruhe, Germany) followed by PON1 detection as described in Section 4.9.

### 4.11. AE Activity

Ca^2+^-dependent AE activities of serum, apoB-DS, and HDL were determined with a photometric assay using phenyl acetate as substrate as described previously [45]. Briefly, either 1.5 µL serum (1:10 diluted), apoB-DS (1:10 diluted), or 2 µg isolated HDL were added to 200 µL of buffer containing 100 mmol/L Tris, 2 mmol/L CaCl_2_ (pH 8.0), and 1 mmol/L phenylacetate. The rate of hydrolysis of phenylacetate was monitored by an increase of absorbance at 270 nm every 30 s. Activities were calculated from the slopes of the kinetic chart.

### 4.12. Cholesterol Efflux Capacity

The HDL cholesterol efflux capacity of HDL (50 µg isolated HDL protein) was assessed in J774 macrophages pretreated with 0.3 mmol/L 8-(4-chlorophenylthio)-cAMP no need to be defined (Sigma-Aldrich, Darmstadt, Germany) as described [27]. 

### 4.13. Association of EV-HDL and EL-HDL with PON1

Twenty-four hours after infection with PON1-Ad, HepG2 cells were washed and incubated with 100 µg/mL of EV-HDL or EL-HDL in DMEM *w*/*o* fetal calf serum (FCS). Aliquots of the incubation mixtures were collected after indicated time intervals followed by 4–20% non-denaturing gradient-gel electrophoresis, PON1 Western blotting, and measurements of AE activity.

### 4.14. PON1 ELISA

PON1 content of isolated human HDL (1 µg/well) was determined with a human PON1 ELISA kit (Biozol, Eiching, Germany) according to the manufacturer’s instruction. 

### 4.15. EL ELISA

Serum EL concentrations were measured with Endothelial Lipase ELISA (Tecan; #JP27182, IBL International, Hamburg, Germany) as described previously [35]. 

### 4.16. RNA Isolation and Quantitative Real-Time PCR Analysis

RNA isolation from mouse liver, reverse transcription, and real-time PCR were performed as described [27]. Primers for mouse PON1 (fw: GGTGTTGGCACTTTACAAGAACC; rev: CGCAAAGGGCCACTTTTACTA), mouse apoA-I (fw. AGCTGAACCTGAATCTCCTG; rev CACTTCCTCTAGGTCCTTGT), and mouse cyclophilin A (fw:CCATCCAGCCATTCAGTCTT; rev: TTCCAGGATTCATGTGCCAG) were from Life Technologies (Vienna, Austria) and for human EL we used Primer Assay QT00078969 (Qiagen, Hilden, Germany).

### 4.17. Lipid and (Apo)Lipoprotein Profiling by Nuclear Magnetic Resonance (NMR) Spectroscopy

Blood serum lipoproteins were measured on a Bruker 600 MHz Avance Neo NMR spectrometer using the Bruker IVDr lipoprotein subclass analysis protocol. Serum samples were thawed, and 330 µL of each sample mixed with 330 µL of Bruker serum buffer (Bruker, Rheinstetten, Germany). The samples were mixed gently, and 600 µL of the mixed sample were transferred into a 5 mm SampleJet rack tube (Bruker). Proton spectra were obtained at a constant temperature of 310 K using a standard nuclear Overhauser effect spectroscopy (NOESY) pulse sequence (Bruker: noesygppr1d), a Carr–Purcedll–Meiboom–Gill (CPMG) pulse sequence with presaturation during the relaxation delay (Bruker: cpmgpr1d) to achieve water suppression, and a standard 2D J-resolved (JRES) pulse sequence (Bruker: jresgpprqf). Data analysis was carried out using the Bruker IVDr LIpoprotein Subclass Analysis (B.I.LISA^TM^, Bruker, Rheinstetten, Germany) method.

### 4.18. HDL-Lipidome Analysis by Liquid Chromatography Coupled to Tandem Mass Spectrometry (LC-MS/MS)

HDL total lipids were extracted according to a modified Bligh and Dyer method: 50 µL of isolated HDL equivalent to 50 µg protein were supplemented with a mixture of internal standards, and lipids were extracted with 1.2 mL methanol/chloroform (2:1, *v*/*v*) in the presence of the antioxidant butylated hydroxytoluene and 310 µL HCL (0.01 N). Phase separation was triggered by the addition of 400 µL CHCl_3_ and 400 µL water. Lipids were quantified by LC-ESI/MS/MS using a Prominence UFLC (Shimadzu, Tokyo, Japan) and QTrap 4000 mass spectrometer (AB Sciex, Framingham, MA, USA). Quantification of phospholipids and sphingolipids was performed in positive-ion mode. A sample (4 µL) was injected to a Kinetex HILIC 2.6 µm 2.1 × 150 mm column (Phenomenex, CA, USA). Mobile phases consisted of water and acetonitrile containing ammonium acetate and acetic acid. The quantification of neutral lipids (triacylglycerols (TAG), diacylglycerols (DAG), cholesteryl ester (CE), and free cholesterol) was performed in positive-ion mode. A sample (4 µL) was injected into an Ascentis Express C18 column (150 × 2.1 mm, 2.7 µm) Sigma-Aldrich, Merck Group). Mobile phases consisted of (A) acetonitrile (ACN):H_2_O = (60:40 *v*/*v*) and (B) isopropanol (IPA)/ACN (90:10 *v*/*v*) supplemented with 10 mmol/L ammonium formate and 0.1% formic acid. Lipid species were detected using scheduled multiple reaction monitoring. 

### 4.19. Statistical Analyses

Data of in vitro and mouse experiments are represented as the means ± standard error of the mean (S.E.M.). Differences between EL and EV samples were assessed by a two-tailed unpaired *t*-test between EL-HDL and EV-HDL or one-way ANOVA when comparing more than two groups (FPLC-derived EV-HDL and EL-HDL fractions) using Graph Pad Prism 5.0. Statistically significant differences between groups are indicated in the plots as follows: *p* < 0.05 (*), *p* < 0.01 (**), or *p* < 0.001 (***). The association between EL and PON1 as well as AE activity in human samples was investigated by correlation analyses using the Spearman correlation coefficient. We performed logistic regression analyses to identify HDL lipid species associated with high vs. low PON1 content, defined via the median, and additionally investigated the influence of EL on significant associations by adjusting these models for high vs. low EL (defined via the median). These analyses were performed using R version 3.6.1.

## Figures and Tables

**Figure 1 ijms-22-00719-f001:**
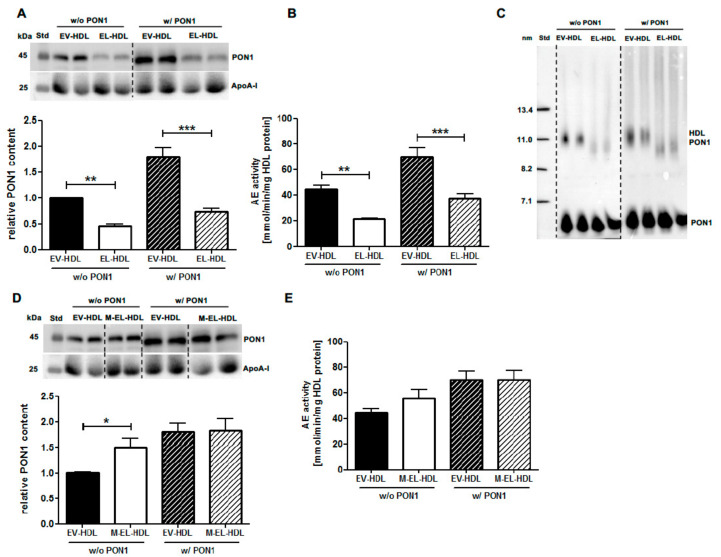
Endothelial lipase (EL) decreases high-density lipoprotein (HDL) paraoxonase 1 (PON1) content and arylesterase (AE) activity. Empty virus (EV)-HDL, EL-HDL, and enzymatically inactive (M)-EL-HDL were generated by incubation of human HDL with EV, EL, or M-EL overexpressing cells in the absence (*w*/*o*) or presence (*w*/) of PON1 overexpression under cell culture conditions for 16 h. Modified HDL was purified by ultracentrifugation. (**A**) First, 10 µg HDL protein were separated by 12% SDS-PAGE followed by Western blotting analyses of PON1 and apolipoprotein A-I (apoA-I). (**B**) PON1 AE activity was measured in 2 µg HDL. (**C**) Aliquots (containing 15 µg protein) were electrophoresed on 4–20% non-denaturing polyacrylamide gels followed by Western blotting analyses of PON1. (**D**) Analyses were performed as in (**A**). (**E**) Analyses were performed as in (**B**). Densitometric values of PON1 in (**A**) and (**D**) were normalized to the corresponding apoA-I signals. Results obtained by densitometry and measurements of AE activity are presented as means + SEM of four independent modifications of human HDL each loaded onto gels (Western blotting) or measured (AE activity) in duplicate. The difference between EL-HDL and EV-HDL or M-EL-HDL and EV-HDL was analyzed by two-tailed unpaired *t*-test in samples *w*/ and *w*/*o* PON1 separately. * *p* < 0.05, ** *p* < 0.01, ****p* < 0.001.

**Figure 2 ijms-22-00719-f002:**
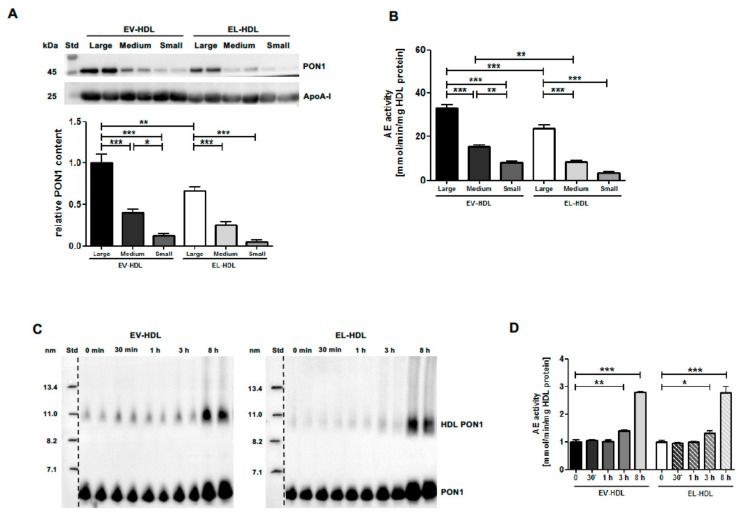
EL decreases HDL size and PON1 content but not the capacity of HDL to associate with PON1. (**A**,**B**) EV-HDL and EL-HDL were fractionated by fast protein liquid chromatography (FPLC) to obtain small, medium, and large EV-HDL and EL-HDL fractions. (**A**) 10 µg HDL protein were separated by 12% SDS-PAGE followed by Western blotting analyses of PON1 and apoA-I. Densitometric values of PON1 were normalized to the corresponding apoA-I signals. (**B**) PON1 AE activity was measured in 2 µg HDL protein. (**C**,**D**) Twenty-four hours after infection with PON1-Ad, HepG2 cells were washed and incubated with 100 µg/mL of EV-HDL protein or EL-HDL protein in Dulbecco’s modified Eagle medium (DMEM). Aliquots were collected after indicated time intervals followed by (**C**) 4–20% non-denaturing gradient-gel electrophoresis of 10 µg HDL protein and subsequent PON-1 Western blotting, and (**D**) measurements of AE activity in 2 µg HDL protein. Densitometry results and AE activity are presented as means + SEM of three (**A**,**B**) or two (**C**,**D**) independent modifications of human HDL each loaded onto gels (Western blotting) or measured (AE activity) in duplicate. The differences between FPLC-derived HDL fractions were analyzed by one-way ANOVA followed by Bonferroni post-hoc test, and the differences between 3 h or 8 h and 0 h time points were analyzed by two-tailed unpaired *t*-test. * *p* < 0.05, ** *p* < 0.01, *** *p* < 0.001.

**Figure 3 ijms-22-00719-f003:**
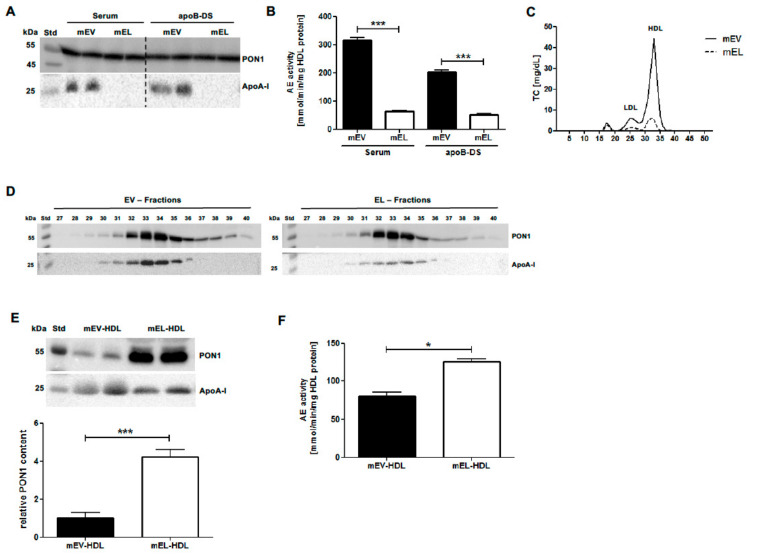
EL overexpression in mice increases the PON1 content and AE activity of HDL. Mouse (m) serum was isolated from blood of EV- and EL-overexpressing mice 48 h after adenovirus (Ad) injection. (**A**) Serum (1.5 µL) and apolipoprotein B (apoB)-depleted serum (apoB-DS) (2 µL), were separated by 12% SDS-PAGE followed by Western blotting analyses of PON1 and apoA-I. (**B**) The AE activity was measured using 1.5 µL of 10-fold diluted serum or apoB-DS. (**C**) Total cholesterol concentrations in the FPLC fractions of mouse serum. (**D**) FPLC fractions 27–40 (10 µL) and (**E**) HDL isolated from serum by ultracentrifugation (10 µg) were analyzed as in (**A**). (**F**) The AE activity using 2 µg HDL protein isolated by ultracentrifugation was measured as described in (**B**). Densitometry and AE activity results are presented as mean + SEM of three independent in vivo modifications, with 12 EL-Ad and four EV-Ad-injected mice per modification. Samples from each modification were loaded onto gels in duplicate or used for AE activity measurements in triplicate. The difference between EL-HDL and EV-HDL was analyzed by two-tailed unpaired *t*-test. * *p* < 0.05, *** *p* < 0.001.

**Figure 4 ijms-22-00719-f004:**
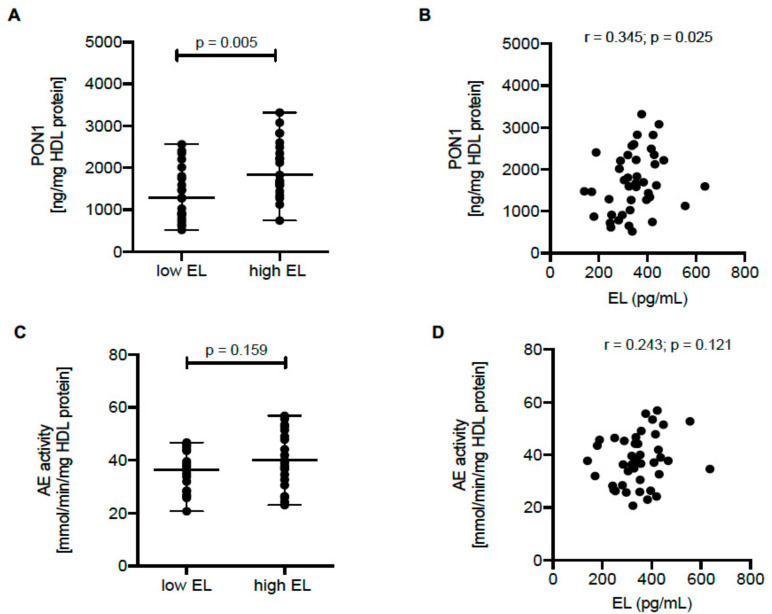
Positive association between HDL PON1 content and EL serum levels in humans. (**A**) The differences in HDL PON1 content (measured by ELISA) between HDL isolated from subjects with high (above median; n = 21) or low (below median; n = 21) EL serum levels were analyzed by two-tailed unpaired *t*-test. (**B**) Correlation of EL serum levels with HDL PON1 content was determined by the Spearman correlation coefficient. (**C**) The differences in AE activity were analyzed as in (**A**). (**D**) Correlation of EL serum levels and AE activity was determined as in (**B**).

## Data Availability

Data is contained within the article or Appendix A.

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
