# Peer review of "Endothelial Lipase Modulates Paraoxonase 1 Content and Arylesterase Activity of HDL"

_ijms, 2021, doi:10.3390/ijms22020719_

Round 1

Reviewer 1 Report

  • The in vitro studies of this manuscript demonstrated an inhibitory effect of endothelial lipase (EL) on the HDL paraoxonase 1 (PON1) content. In contrast, overexpression of EL in mice increased serum HDL PON1 content. Reduction in serum HDL level was suggested as a mechanism for the increased HDL PON1 content. The serum apoAI level was dramatically reduced in EL-ad injection mice. Does EL inhibit apoAI expression? Was the EL transgene expressed in hepatocytes or other cell types, such as endothelial cells? What promoter was associated with the EL transgene?
  • Determination of the EL expression level in the EL-ad-injected mice is necessary. Accumulation of newly synthesized EL in cells may inhibit global protein synthesis, including apoAI, if it occurs hepatocytes.
  • Do the low-EL subjects have low level of HDL and/or apoAI? Namely, is low serum HDL/apoAI level a mechanism for the high HDL PON1 content as shown in EL-ad injected mice?
  • The figure legends are difficult to be followed. Can they be reorganized following the alphabet order as shown in the figures?
  • Fig 2A: same amount of HDL protein (10µg) was loaded for immunoblot; however, the apoAI blot density appears to be weaker in the EL-HDL samples than the EV-HDL samples, why?
  • Fig 2A and 2B: are the difference of the PON1 level/AE activity between the small and medium EL-HDL not statistically significant?

Reviewer 2 Report

Regarding the manuscript entitled “Endothelial Lipase Modulates Paraoxonase 1 content and arylesterase activity of HDL” the authors present an extremely complete approach with in vitro and in vivo assays (mice and human).

The introduction is well structured and present the problem in a clear way. The results are fine organized and presented. The discussion is well structured and authors assume clearly the differences obtained between in vitro and in vivo assays, and present some probably explanations for the obtained results.

The conclusions are consistent with the data in the results, however, should be highlighted the need of continuing developing this topic with in vivo assays involving a higher number of participants in order to clarify the role of endothelial lipase.

Round 2

Reviewer 1 Report

As the author described, previous studies reported a negative correlation between EL and HDL serum levels in patients with cardiovascular disease but not in those without. In line, the EL serum levels in the present study were not significantly correlated with either parameter indicative of HDL. Please add this finding in the abstract, such as: ‘In healthy subjects, the serum EL level was not significantly correlated with HDL level. However, the HDL PON1 content was positively associated with the EL serum level.’

Author Response

Reply to reviewer 1.

We would like to thank the reviewer for the suggestion regarding revision of the Abstract section.

As the author described, previous studies reported a negative correlation between EL and HDL serum levels in patients with cardiovascular disease but not in those without. In line, the EL serum levels in the present study were not significantly correlated with either parameter indicative of HDL. Please add this finding in the abstract, such as: ‘In healthy subjects, the serum EL level was not significantly correlated with HDL level. However, the HDL PON1 content was positively associated with the EL serum level.’

We have followed the reviewer’s suggestion and added new information in the abstract as follows:

‘’ In healthy subjects, EL serum levels were not significantly correlated with HDL levels. However, HDL PON1 content was positively associated with EL serum levels.’’